# Synthesis of Multicolor Carbon Dots Based on Solvent Control and Its Application in the Detection of Crystal Violet

**DOI:** 10.3390/nano9111556

**Published:** 2019-11-01

**Authors:** Dan Zhao, Xuemei Liu, Zhixia Zhang, Rui Zhang, Liangxiu Liao, Xincai Xiao, Han Cheng

**Affiliations:** School of Pharmaceutical Sciences, South-Central University for Nationalities, Wuhan 430074, China; lxmhbs@163.com (X.L.); zhixia_zhang@163.com (Z.Z.); zraixbl@163.com (R.Z.); Chebyshev7wait7llx@163.com (L.L.); xcxiao@126.com (X.X.); Chenghan@mail.scuec.edu.cn (H.C.)

**Keywords:** carbon dots, multicolor luminescence, solvent control, surface state, crystal violet sensing

## Abstract

The adjustment of the emitting wavelength of carbon dots (CDs) is usually realized by changing the raw materials, reaction temperature, or time. This paper reported the effective synthesis of multicolor photoluminescent CDs only by changing the solvent in a one-step solvothermal method, with 1,2,4,5-tetraaminobenzene as both the novel carbon source and nitrogen source. The emission wavelengths of the as-prepared CDs ranged from 527 to 605 nm, with quantum yields (QYs) reaching 10.0% to 47.6%, and it was successfully employed as fluorescence ink. The prepared red-emitting CDs (R-CDs, λ_em_ = 605 nm) and yellow-emitting CDs (Y-CDs, λ_em_ = 543 nm) were compared through multiple characterization methods, and their luminescence mechanism was studied. It was discovered that the large particle size, the existence of graphite Ns, and oxygen-containing functional groups are beneficial to the formation of long wavelength-emitting CDs. Y-CDs responded to crystal violet, and its fluorescence could be quenched. This phenomenon was thus employed to develop a detection method for crystal violet with a linear range from 0.1 to 11 µM and a detection limit of 20 nM.

## 1. Introduction

Carbon dots (CDs), as a novel carbon nanomaterial, have been widely applied in the fields of detection [1,2], drug delivery [3,4], medical diagnosis [5,6], and biological imaging [7] due to its excellent fluorescence properties, low cell toxicity, good biological compatibility, and low cost [8,9,10]. Recently, various CDs with different luminesces have been developed by researchers. In the present published literature, an adjustment of the emission wavelength of prepared CDs was realized mainly through the adjustment of raw materials and their amounts [11,12], the reaction time or temperature [13], and changing the solvents [14]. The adjustment of raw materials is often the most commonly used method. Lin et al. [11] prepared red-, orange-, yellow-, and green-emissive CDs by changing the raw material type (p-phenylenediamine or o-phenylenediamine) and their amounts. Chen et al. [13] realized surface state control by adjusting the reaction time or temperature, and finally acquired blue- and orange-emissive CDs (p-phenylenediamine as the raw material, solvothermal route). However, the type of as-prepared CDs through this method is quite limited. In recent years, some research groups discovered that adjusting the solvent is also effective in preparing long wavelength-emissive CDs. Xiong et al. [14] acquired all color-emissive CDs (λem ranged from 443 to 745 nm) by adjusting the type of solvent (formamide, *N*,*N*-dimethylformamide, ethanol, and concentrated sulfuric acid) and their ratios (L-glutamic acid and *o*-phenylenediamine as raw materials). This route also exhibited obvious drawbacks, such as a high reaction temperature (210 °C), long reaction time (10 h), and the employment of a strong acid. Tian [15] and his group also realized all color-emissive CDs through three different solvents (water, glycerin, and *N*,*N*-dimethylformamide), with citric acid and urea as raw materials. However, this method requires careful control of the solvent ratios. These reports offered a new and excellent method for the research in this field, but the relevant reports on the synthesis based on single raw materials have rarely been seen in published literature. Therefore, studies on the impacts of solvents upon CD synthesis are of great significance.

An in-depth insight into the formation mechanism of CDs is of great significance not only for understanding the construction chemistry of carbon nanomaterials but also for guiding large-scale synthesis. To study the synthesis mechanism, Xiong et al. [14] realized the adjustment of CDs’ emission wavelength using multiple solvents. The particle size and extent of graphitization of CDs greatly depend upon the condition of dehydration and carbonization. Because of the catalysis effect of concentrated sulfuric acid, the accelerated dehydration and carbonization process produced near-infrared-emissive CDs with the largest particle size and highest graphite nitrogen content. Some researchers realized the adjustment of the emission wavelength of as-prepared CDs by dissolving CDs into different solvents. For instance, Chao et al. [16] studied the impact of solvents upon the optical properties of CDs by dissolving CDs (prepared by o-phenylenediamine) into different solvents. Wang [17] and his team proposed that the absorption and photoluminescence of CDs in alcoholic solvents are mainly determined by the –OH groups of solvents rather than their polarity. In the aspect of the luminescence mechanism, Xiong et al. [14], through detailed characterization, proved that the differences in the particle sizes and element contents of CDs cause varied fluorescence emission properties. Because the process of CD synthesis is difficult to monitor, especially for solvent methods, and the types of solvents are too many and their characteristics vary greatly, exploration of the synthesis and luminescence mechanism of CDs through solvent methods is hard but meaningful work.

Crystal violet (CV) is a triphenylmethane dye used to treat infections. It is an antifungal agent and has been used in the treatment of aquatic animal diseases and as a water disinfectant [18]. CV is highly toxic, and may be carcinogenesis and mutagenesis to humans [19,20]. Therefore, the development of sensitive detection methods to CV is of great importance. Some researchers have applied capillary electrophoresis [21], liquid chromatography-mass spectrometry [22], and Raman spectroscopy [23] in crystal violet detection. However, these methods require cumbersome pre-processing and are time consuming, so the exploration of a fast, simple, and sensitive method is necessary.

This paper reports the simple and fast one-step synthesis of multiple-color-emissive CDs through the solvothermal route with a novel raw material, 1,2,4,5-tetraaminobenzene, as the single carbon source. The effects of solvents on the synthesis and luminescence mechanism of CDs were investigated by various characterizations. The properties of the solvents had an important influence on the synthesis process while the particle size, element form, and their content caused a change of the emission wavelength. The prepared yellow-emissive CDs were further applied as an effective fluorescence sensing platform for CV, with excellent selectivity and sensitivity.

## 2. Experimental Section

### 2.1. Reagents and Materials

1,2,4,5-tetraaminobenzene (analytically pure, 95%) was purchased from Adamas Reagent Co. Ltd. (Shanghai, China).; organic solvents (analytical grade) were purchased from Sinopharm Chemical Reagent Co. Ltd. (Shanghai, China).; and Rhodamine 6G and Rhodamine B were purchased from Aladdin Chemistry Co. Ltd. (Shanghai, China). The MCI Gel filler (CHP20/P120) was purchased from Mitsubishi Chemical Corporation of Japan (Tokyo, Japan). Crystal violet (CV), methyl red (MR), SudanI (SDI), methylene blue (MB), malachite green (MG), FeCl_3_, HgCl_2_, (CH_3_COO)_2_Pb, AgNO_3_, CdCl_2_, CoCl_2_, CuCl_2_, ZnCl_2_, and CaCl_2_ were purchased from Sinopharm Chemical Reagent Co. Ltd. (Shanghai, China).

### 2.2. Instrumentations

A Lambda-35 UV-vis spectrophotometer and LS55 fluorescence spectrophotometer were purchased from PerkinElmer; a VG Multilab 2000 X-ray photoelectron spectroscopy and Nicolet 6700 Fourier infrared spectrometer were purchased from Thermo Fisher Scientific (KBr tableting method, Waltham, MA, USA). TEM images were obtained using a JEM-1400Plus transmission electron microscope (Japan Electron Optics Laboratory Company, Tokyo, Japan). X-ray power diffraction (XRD) patterns were recorded on a D8-Advance spectrometer (Bruker, Germany). The Raman spectra were recorded using a Raman spectrometer from Thermo Fisher Scientific (Waltham, MA, USA).

### 2.3. Preparation of CDs

CDs were synthesized through the one-step solvothermal method. In total, 0.025 g of 1,2,4,5-tetraamino-benzene powder were placed in a 25-mL round-bottomed flask. Next, 24 mL of solvent were added to the flask, and stirred for 10 min under the protection of N_2_ until totally dissolved. Then, 24 mL of the solution were then transferred into a stainless-steel autoclave with Teflon lining, which was then placed into an oven at 170 °C for 5.5 h. The autoclave was naturally cooled to room temperature, and the CDs solvent was acquired. The prepared CDs solvent was filtered through 0.22-µm filter, and then after the ultrafiltration purification process, the solvent was dried into solids. MCI Gel was used as the filler for the chromatographic column (separation mechanisms include a molecular sieve and polarity), with methanol and water as the eluents. The prepared R-CDs and Y-CDs were added into the column to be separated. Then, different separated components were obtained for characterization.

### 2.4. Quantum Yield (QY) Measurement

Rhodamine 6G (QY = 95%) and rhodamine B (QY = 56%) dissolved in absolute ethanol (refractive index η = 1.096, 25 °C) were used as references, respectively [12]. The QYs of CDs were calculated by comparing the ratio of the fluorescence area (rhodamine 6G-λex = 488 nm and rhodamine B-λex = 495 nm) to the absorbance. All samples were dissolved in absolute ethanol and their absorbance at 488 or 495 nm was controlled to be less than 0.1. The relative QY was calculated as follows:ΦX = ΦST (GradX/GradST)(ηX^2^/ηST^2^),
where Φ is QY, Grad is the ratio of the fluorescence area to the absorbance, η is the refractive index of the solvent, ST is the standard substance, and X is the CDs sample, where ηX = ηST.

### 2.5. Fluorescence Detection of Crystal Violet

The total volume of Y-CDs solution, different detected samples, and ethanol was 1 mL, followed by the addition of CV standard with various concentrations. The fluorescence spectra were recorded after reaction for 1 min at room temperature. The selectivity of CV sensing was confirmed by adding metal ion solutions (including Fe^3+^, Hg^2+^, Pb^2+^, Ag^+^, Cd^2+^, Co^2+^, Cu^2+^, Zn^2+^, and Ca^2+^ ions) and other dyes (MR, SudanI, MB, and MG) instead of CV. The fluorescence was recorded at the excitation wavelength of 500 nm.

## 3. Results and Discussion

### 3.1. The Preparation of Multiple-Color-Emissive CDs Based on Different Solvents

1,2,4,5-tetraaminobenzene was selected as the single raw material to prepare CDs through the solvothermal route. With the other synthesis parameters fixed, the change of solvents would cause an obvious difference in the emission wavelength of the as-prepared CDs. To systematically investigate this phenomenon, several solvents, including water, absolute ethanol (EA), isopropanol (IPA), methanol (MT), *n*-butanol (NBA), and *N*,*N*-dimethylformamide (DMF), were selected to prepare CDs under the same synthesis environment. Interestingly, the CDs prepared with water as the solvent had weak fluorescence. Other CDs prepared by organic solvents emitted five different colored fluorescence, and their fluorescence spectra, UV-vis spectra, excitation wavelengths, emission wavelengths, full width at half maximum (FWHW), QYs, and other related data are shown in Figure 1a and Table 1. Their excitation wavelengths were in the range of 460 to 540 nm, emission wavelengths of 527 to 605 nm, and FWHW of 44 to 86 nm. The absorption peaks of the UV-vis spectra of the five samples were different, illustrating an obvious structure difference among the samples. The ones with longer UV-vis absorption wavelengths exhibited longer emission wavelengths. The prepared multi-colored CDs have great potential as fluorescence ink for fluorescence color painting (Figure 1b).

It is regarded that properties of the solvent, such as the dehydrating ability, solubility, boiling point, polarity, protic, and non-protic property, lead to differences in the aggregation, dehydration, carbonization, core formation, and growth process during synthesis, and thus differences in the emission wavelengths of the as-prepared CDs.

First, the dehydrating ability of the solvent plays a critical role in the as-prepared CDs, while in the aqueous phase, the reactants can hardly aggregate through the dehydration reaction. The CDs solution prepared with water as the solvent is brown, showing that the high temperature already initiates the carbonization process of the raw material, but the whole reaction environment is not beneficial for the formation of nano-particles through aggregation of the raw material by the dehydration process. Therefore, the prepared solution basically emits weak fluorescence. Comparatively, the organic solvents offer a better environment for the dehydrating process. Some researchers reported [24] that the organic solvothermal method is more beneficial for the preparation of long wavelength-emissive CDs, and thus more types of CDs.

Then, the solubility of raw materials in solvents also influences the properties of CDs, and is also an important factor for the filtration of solvents. A series of solvents, including water, absolute ethanol, isopropanol, methanol, *n*-butanol, *N*,*N*-dimethylformamide, formamide, *N*,*N*-dimethyl sulfoxide, tetrahydrofuran, and cyclohexane, were investigated regarding their impacts on the as-prepared CDs. The experiments showed that the carbon source exhibited excellent solubility in alcoholic solutions and *N*,*N*-dimethylformamide. The low solubility of the raw material would lead to an aggregation state in the solvent with low dispersion, which is unbeneficial to the formation of small-sized nano particles.

Meanwhile, the boiling point of the solvent would also impact the preparation of CDs. The solvothermal route seals solvents of the same volume into the 25-mL lining of the reaction kettle under heating. Under the same reaction space and reaction temperature, the difference in boiling points of the solvents would cause a difference in the steam pressure of the environment. Among the tested solvents, ethanol, isopropanol, and methanol possess lower boiling points, and thus higher steam pressure in the autoclave, which is more beneficial to the growth of long wavelength- emissive CDs. For *n*-butanol, its high boiling point (117 °C) causes a lower steam pressure, and shorter wavelength-emissive CDs (yellow fluorescence). DMF possesses the highest boiling point (153 °C), and therefore the lowest reaction pressure, so the as-prepared CDs exhibit the shortest emission wavelength. The experiment results are in accordance with the reports of Xiong’s team [14], who found that a lower boiling point of the solvent could improve the dehydration and carbonization process. Therefore, in the solvothermal route, the selection of a solvent with a proper boiling point is another way to realize the preparation of CDs with different colored fluorescence.

What needs to be emphasized is that the polarity difference of solvents renders an obvious difference in the interaction between the solvent and carbon source, leading to an optical property change of the as-prepared CDs. First, the differences in the UV-vis spectra of the solvents prove our inference, as shown in Figure 2. The nitrogen in the amino group of 1,2,4,5-tetraaminobenzene possesses lone pair electrons, and the molecules have a benzene ring, with UV absorption in the form of the n–π* transition. According to the absorbance spectroscopy theory, with the increase of the solvent polarity, the n electrons of the solvent molecules form a hydrogen bond with the polar solvent in the n–π* transition, which decreases the energy in the n track and enhances the energy difference between the n and π* track, leading to the blue shifts of λ_max_ of the absorption band. As shown in Figure 2, the concentration of the sample to be tested was 1 mg/mL. In the DMF solution (which has the strongest polarity), its UV-vis spectrum exhibits an absorption peak at 450, 483, and 523 nm; in the medium polarity solutions (MT and EA), the absorption peaks in MT are at 479, 518, and 621 nm, and in EA the peaks are at 479, 518, and 633 nm. The result proves that the increase of the solvent polarity causes the λ_max_ of the absorption band blue shift. The UV-vis spectra of DMF, MT, and EA are discrete, while in isopropanol and *n*-butanol, whose polarities are smaller, the absorption spectra are wide and integrated, with peaks at 505 and 463 nm.

The difference between solvents is not only their polarity but also their protic (non-protic) property. In our experiments, the alcohol solutions belong to polar protic solutions, which is more beneficial to the preparation of long wavelength-emissive CDs, while DMF is a polar non-protic solution, and the CDs as-prepared with it would have shorter emission wavelengths. The experiment results and related literature reports [16] showed that the properties of the solvents, including their dehydrating ability, solubility, boiling point, and polarity, are critical factors to the properties of as-prepared CDs. Of course, some other factors might have been neglected, which can be investigated by other raw materials in our future research. The properties of the solvent (solubility, boiling point, polarity, and protic (non-protic)) are important factors affecting the preparation of CDs.

Properties of the solvents will not only impact the synthesis process of CDs but also the optical properties of prepared CDs. To investigate the influence of solvents upon the optical properties of prepared CDs, the fluorescence and UV-vis properties of red-emissive CDs (R-CDs, λ_max_ = 605 nm) were investigated. R-CDs were dissolved into the solvents, including water, methanol (MT), absolute ethanol (EA), isopropanol (IPA), acetone (CP), ethyl acetate (EAC), and dichloromethane (MC). The acquired emission wavelengths, fluorescence intensities, and UV-vis spectra are shown in Figure 3. The experiment results showed that the emission of CDs is highly dependent upon the polarity and type of the solvents. As shown in Figure 3a,b, for protic solvents, the CDs dissolved in water exhibit the longest emission wavelength (615 nm), and in some alcoholic protic solvents, like MT, EA, and IPA, the emission wavelengths of CDs are almost the same (about 607 nm). The solvents also influence the fluorescence intensity of dissolved CDs. The fluorescence intensity of CDs is the weakest in water but the strongest in ethanol. The absorption and photoluminescence of CDs in alcoholic solvents mainly originate from the –OH group, rather than the polarity of the solvents, which explains why CDs exhibit the strongest fluorescence intensity in ethanol [17]. As shown in Figure 3c, with the polarity decrease of protic solvents (MT, EA, and IPA), the absorption peak of R-CDs red shift. The solvent-dependency of R-CDs originates from the hydrogen bond interaction between the protic solution and CDs [25,26,27]. For non-protic solvents (CP, EAC, and MC), the polarity decrease would cause red shifts in the emission wavelength of dissolved CDs (Figure 3a), and the UV-vis absorption wavelength blue shifts with the increase of the polarity (Figure 3c). In different non-protic solvents, dipole-dipole interaction plays a critical role in PL shifting, which is in accordance with the related reports [25,26,27].

The above studies show that the dehydrating ability, solubility, boiling point, polarity, and (non-) proton of solvents have a great impact on the synthesis process of CDs, leading to obvious differences in the optical properties of prepared CDs. The dissolution of CDs into different solvents also changes the fluorescence intensity and emission wavelength. This solvent dependency of CDs is helpful for the effective adjustment of the optical properties of CDs. 

Some means of characterization were used to characterize the CDs as-prepared with these two solvents (ethanol and *n*-butanol), and their differences in terms of particle size and surface component were further compared.

The fluorescence spectrum of R-CDs and Y-CDs were compared. As shown in Figure 4a,b, R-CDs shows a weak excitation wavelength dependency while Y-CDs exhibit a strong dependency. MCI Gel column chromatographic separation was used to study the dependency of both R-CDs and Y-CDs. After the separation of R-CDs, three different emissive CDs were acquired with emission wavelengths at 619, 608, and 554 nm (Appendix A) while for Y-CDs, four samples were acquired with emission wavelengths at 476, 484, 543, and 586 nm (Figure 4c). Although both original CDs can be further separated into different color-emissive CDs, the emission wavelengths of those CDs acquired from R-CDs exhibit a 65-nm difference, while those of CDs acquired from Y-CDs exhibit a 110-nm difference. Both of the prepared CDs (R-CDs and Y-CDs) are heterogeneous samples, but Y-CDs were separated into more samples, and thus exhibit a stronger emission wavelength dependency. Furthermore, when choosing different excitation wavelengths, those nano particles with different particle sizes play different roles in illumination. The CDs with a smaller particle size and larger surface defect energy gap take a leading role in illumination under the excitation of shorter wavelength, while with the increase of the excitation wavelength, those with narrower surface defect energy gap gradually take the leading roles. The CDs prepared by different solvents contain nanoparticles with a large range of particle sizes, and their surface defect energy gaps also show great differences, which results in the exhibition of an excitation wavelength dependency. The quantum confinement effect, and the surface state is similar to the molecule state, both of which lead to the complexity of the excitation state of CDs [28,29,30].

TEM was used to characterize the morphology and particle size of prepared R-CDs and Y-CDs. As shown in Figure 5, both samples exhibit a spherical shape with an average particle size of 6.99 nm for R-CDs and 5.32 nm for Y-CDs. The difference in particle size might originate from the difference of the solvents used during the preparation process. Because of the impact of the quantum confinement effect, the nanoparticle with a larger particle size would be more beneficial to the formation of long wavelength-emissive CDs. The XRD patterns of R-CDs and Y-CDs (Appendix A) show highly disordered carbon atoms [31].

FTIR, Raman, and XPS spectra were used to investigate the surface functional groups, element compositions, and existence forms of R-CDs and Y-CDs. As shown in Figure 6a, FTIR spectra show the both R-CDs and Y-CDs possess rich functional groups on their surfaces. R-CDs and Y-CDs exhibit absorption peaks at 3141, 3128 (N–H), 1623 (–CONH–), 1587 (C=O), 1503, 1493 (C=C), 1404, 1408 (C–H), 1305, 1340 (C–N), 1018, and 1233 cm^−1^ (C–O). The absorption peaks at 841 and 672 cm^−1^ of both CDs represent the bending vibration of C–H on the benzene ring. The peak at 3046 cm^−1^ in R-CDs’ spectra represents the stretching vibration of =CH, while the peaks at 2965, 2933, and 2878 cm^−1^ in Y-CDs’ spectra represent the stretching vibration of C–H, and peaks at 2832 and 2767 cm^−1^ represent the existence of the O=C–H group [17,28]. The absorption peaks in Y-CDs’ FTIR spectra are much more complicated than those in R-CDs’, further proving that the Y-CDs prepared with *n*-butanol complicated the surface structure, the same as the conclusion obtained from the analysis of the CDs’ excitation wavelength dependency. Moreover, the Raman spectra of the two peaks at 1363 and 1551 cm^−1^ correspond to the disordered structures or defects (D band) and the graphitic carbon domains (G band) (Appendix A). The intensity ratios of I_D_/I_G_ are 1.14 and 1.04 for R-CDs and Y-CDs, respectively, indicating that R-CDs are more disordered and amorphous than Y-CDs [32,33].

The XPS full spectra of R-CDs and Y-CDs exhibit three absorption peaks at 248, 400, and 532 eV, which are attributed to C1s, N1s, and O1s [34] (Appendix A). In the spectrum of C1s (Appendix A), R-CDs and Y-CDs possess the same functional groups, showing three absorption peaks at 284.4, 285.2, and 288.4 eV, which are attributed to C=C/C–C, C–N/C–O, and C=O groups. In the spectrum of N1s (Figure 6b,c), R-CDs and Y-CDs show the same absorption peaks at 400.4, 398.4, and 399.4 eV, which are attributed to pyrrole Ns, pyridine Ns, and amino Ns. Interestingly, R-CDs exhibit a peak at 401.4 eV because of the existence of graphite Ns, which Y-CDs does not. In the spectrum of O1s (Appendix A), both CDs exhibit two absorption peaks at 531.7 and 532.3 eV, which are attributed to the existence of C–O and C=O, which is in accordance with the results of the FTIR analysis. The combined types of nitrogen in R-CDs are more than that of Y-CDs, and the oxygen content of R-CDs is also obviously more than that of Y-CDs (Table 2), proving that the surface state of CDs is an important factor to the optical properties of CDs.

As shown in Figure 7, the difference in solvents would lead to an obvious variety in the particle size and surface state of the prepared CDs. Based on the results of multiple characterization methods, the possible luminescence mechanism prepared by different solvents can be proposed. The photoluminescence properties are controlled by the quantum confinement effect and surface state, which is in accordance with the conclusions of most of the literature [35,36]. As discussed in Section 3.1, compared with *n*-butanol, ethanol exhibits better solubility to raw materials, and its low boiling point causes higher reaction pressure in the autoclave under the same reaction temperature and time; its stronger polarity is also beneficial to the formation of products with a larger particle size and more oxygen on the surface. Because of the quantum confinement effect, CDs with a smaller particle size exhibit a shorter emission wavelength, while larger CDs exhibit a longer wavelength, as proven by the TEM images. On the one hand, XPS shows the existence of graphite Ns in R-CDs, which produces a middle gap state inside the gap between the highest occupied molecular orbital (HOMO) and the lowest unoccupied molecular orbital (LUMO) in the undoped system, leading to an obvious red shift absorption and then the production of fluorescence at the low energy end in the visible light spectrum. This is in accordance with the conclusion reported by HoláK [37] that the red-shifting trend of CDs is the result of increased graphite Ns in CDs’ structures. On the other hand, the result of XPS also shows that the content of oxygen in R-CDs is obviously more than that in Y-CDs (Table 2). Because the luminescence center on the surface of CDs is mainly composed of the binding of conjugated atoms with oxygen atoms, the gap between HOMO and LUMO is greatly determined by the doping content of oxygen [27]. It is well known that heteroatom doping has a critical role in the PL properties, but the literature has pointed out that the size of the electronegativity of heteroatoms also affects the emission wavelength of CDs. Yang et al. [38] believe that a blue shift of the photoluminescent emission can be observed by doping with more electronegative elements than C (such as N), and a red shift can be obtained by doping with less electronegative atoms, such as S. For our research, since the electronegativity of oxygen is stronger than that of nitrogen, the reduced doping amount of oxygen atoms would lead to a stronger trend of red-shifting of the emission wavelength of as-prepared CDs than the same amount of nitrogen atoms. Therefore, a large particle size, the existence of graphite Ns, and an increase of the oxygen content are all reasons for the redshift of the emission wavelength of the prepared CDs.

### 3.2. Fluorescent Detection of CV

The selectivity of the Y-CDs nanosensor for CV was investigated. Different samples, including CV, MR, SudanI, MB, MG, Fe^3+^, Hg^2+^, Pb^2+^, Ag^+^, Cd^2+^, Co^2+^, Cu^2+^, Zn^2+^, and Ca^2+^ (concentration: 10 µM), were reacted with Y-CDs solution. As shown in Appendix A, the F_0_/F (F_0_ and F are the fluorescence intensity before and after the addition of detected samples) of the Y-CDs could be visibly quenched by the addition of CV, and there were a slimly change of F_0_/F of the Y-CDs after the addition of metal ions and other dyes. This result showed that the Y-CDs were selective toward CV over the other metal ions, and dyes. The Y-CDs could be developed as an efficient fluorescence sensor for CV. Figure 8 shows the quenching effect of the fluorescence intensity towards CV. The fluorescence intensity of Y-CDs decreases gradually with increasing concentration of CV (the concentration of CV is 0, 0.1, 0.5, 0.9, 1.3, 1.8, 2.5, 3.0, 4.0, 5.0, 6.0, 7.0, 8.0, 9.0, 10.0, and 11.0 µM). The F_0_/F values of the Y-CDs were treated with a concentration gradient of CV. A good linear correlation (R^2^ = 0.99807) was found over the concentration range of 0.1 to 11 µM and the LOD, according to the IUPAC (International union of pure and applied chemistry) standard, which was taken as 3× standard deviation/slope, calculated as 20 nM, which is comparable with the reported data [39]. Han [40] et al. realized the detection of crystal violet in fish tissues based on yellow-emissive silicon nanoparticles, with a detection limit of 25 μg/mL (6.12 × 10^−5^ M), while in our study, the detection limit was 20 nM, lower than that in Chen’s detection.

For the fluorescence quenching mechanism, the UV-vis absorption spectra of CV and the emission spectra of Y-CDs were compared (Appendix A). The UV-vis absorption spectrum of CV shows a broad absorption peak at 510 to 675 nm, which greatly overlaps with the emission spectra of Y-CDs, resulting in CV’s effective screening of the emission of Y-CDs. This indicates that the Y-CDs are quenched by CV due to the internal filtration effect. For the detection methods based on the inner filter effect, the realization of sensitive detection not only relies on the overlapping extent of the fluorescence spectra of CDs with the absorption spectra of crystal violet but also on the distance between the energy donor (Y-CDs) and its acceptor (dye). Besides, crystal violet also shows good solubility in ethanol. Therefore, this method can realize sensitive detection of crystal violet.

## 4. Conclusions

This paper investigated the preparation of CDs with 1,2,4,5-tetraaminobenzene as the raw material. The emission wavelength is adjustable merely through the control of the solvent. Properties of the solvents, including the dehydrating ability, solubility, boiling point, polarity, and (non-) proton, cause the difference in the particle size and surface state of the prepared CDs, which further influence its photoluminescence property. The particle size and element composition and content also impact the emission wavelength of CDs. The emission wavelength of prepared CDs red shifts with a stronger quantum confinement effect, the existence of graphite Ns, and more oxygen-doping on the surface. Furthermore, Y-CDs were applied as a probe for the detection of CV with a linear range of 0.1 to 11 µM and LOD of Y-CDs of 20 nM, with the detection mechanism as the fluorescence inner filter effect.

## Figures and Tables

**Figure 1 nanomaterials-09-01556-f001:**
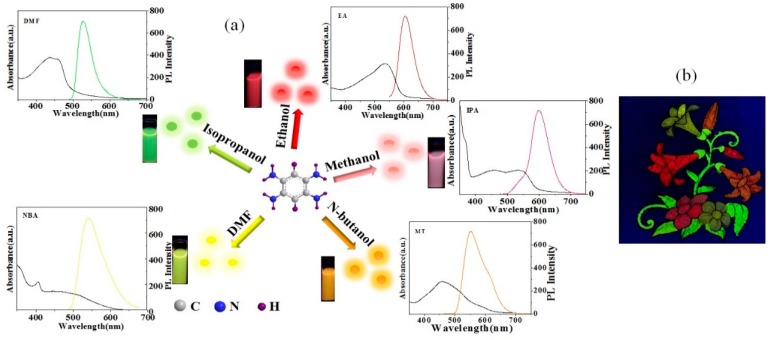
(**a**) The fluorescence and UV-vis spectra of carbon dots (CDs) synthesized by five different solvents. (**b**) A fluorescent ink color photo (365 nm) drawn by five CDs.

**Figure 2 nanomaterials-09-01556-f002:**
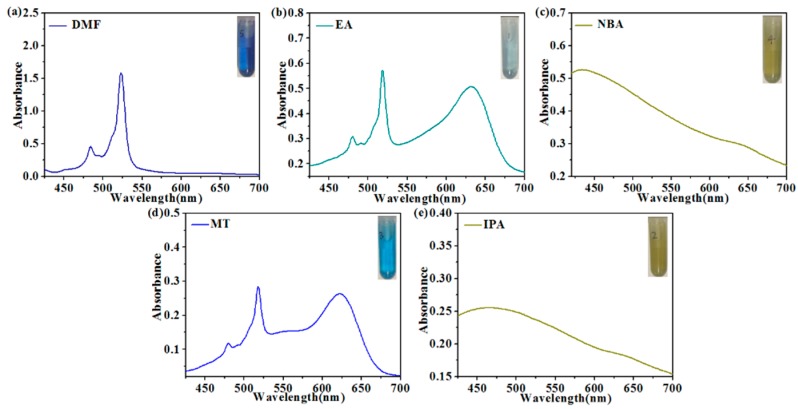
The UV-vis spectrum of the raw material dispersed in *N*,*N*-dimethylformamide (**a**), ethanol (**b**), *n*-butanol (**c**), methanol (**d**), and isopropanol (**e**); the insets are photographs of the solution under ambient light.

**Figure 3 nanomaterials-09-01556-f003:**
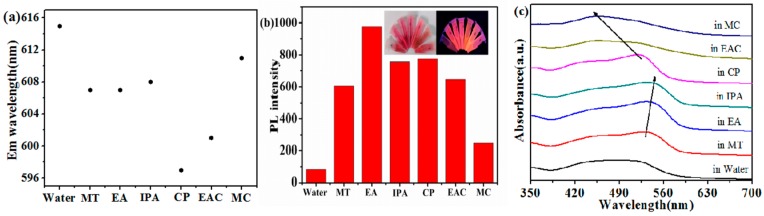
(**a**) Plot of the emission wavelengths of R-CDs dispersed in different polar solvents. (**b**) Bar graph of the fluorescence intensity of R-CDs dispersed in different polar solvents; insets are photographs under 365 nm UV-light (from left to right, water, MT, EA, IPA, CP, EAC, MC). (**c**) UV-vis spectra of R-CDs dispersed in different polar solvents.

**Figure 4 nanomaterials-09-01556-f004:**
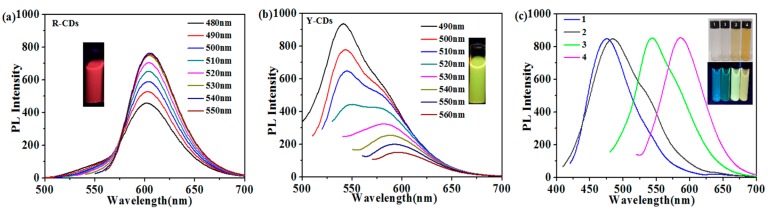
(**a**) Fluorescence spectra of R-CDs under different excitations. (**b**) Fluorescence spectra of Y-CDs under different excitations. The insets are photographs under 365 nm UV lamps. (**c**) Y-CDs fluorescence spectra of four samples obtained by separation. The insets are photographs of the samples under ambient light and 365 nm UV lamps.

**Figure 5 nanomaterials-09-01556-f005:**
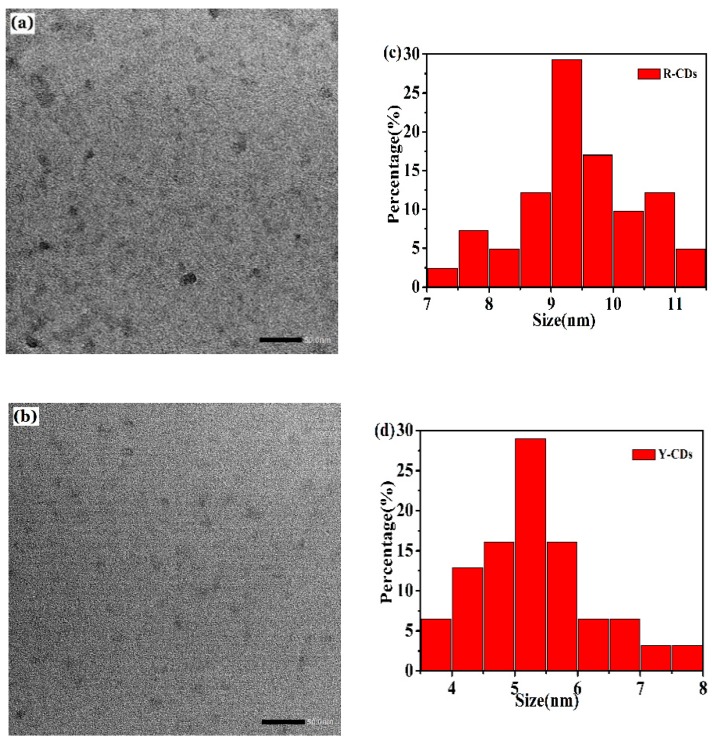
(**a**) and (**b**) are TEM images of R-CDs and Y-CDs, respectively. (**c**) and (**d**) are the particle size distributions of R-CDs and Y-CDs, respectively.

**Figure 6 nanomaterials-09-01556-f006:**
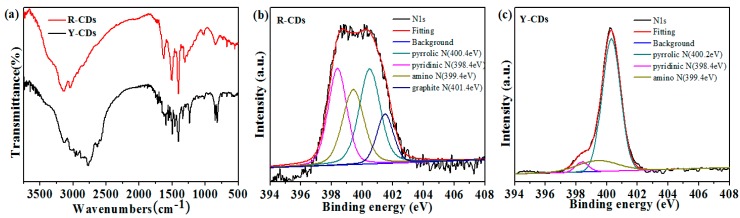
(**a**) FTIR spectrum of R-CDs and Y-CDs. (**b**) N1s spectrum of R-CDs. (**c**) N1s spectrum of Y-CDs.

**Figure 7 nanomaterials-09-01556-f007:**
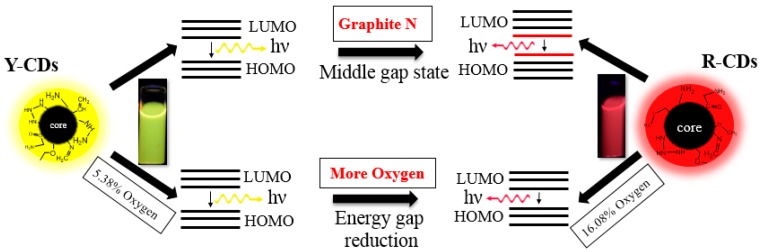
A possible luminescence mechanism for CDs.

**Figure 8 nanomaterials-09-01556-f008:**
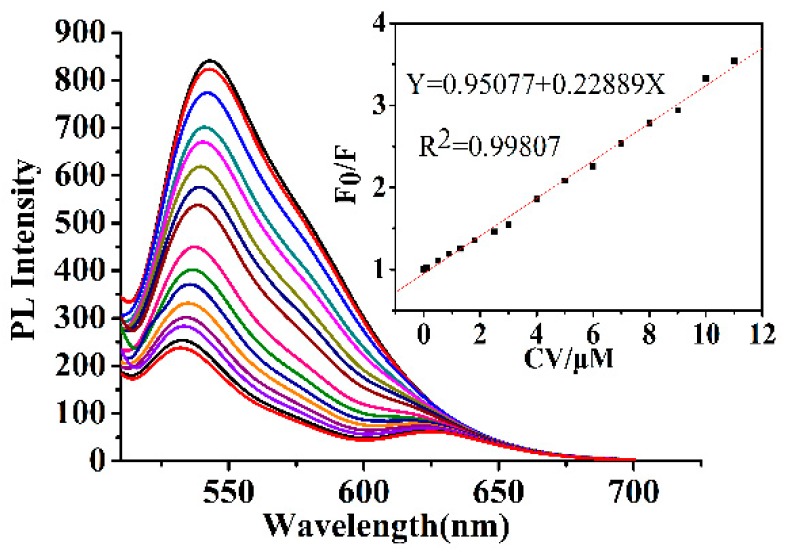
Emission spectra of Y-CDs after the addition of Crystal violet (from top to bottom: [Crystal violet] = 0, 0.1, 0.5, 0.9, 1.3, 1.8, 2.5, 3.0, 4.0, 5.0, 6.0, 7.0, 8.0, 9.0, 10.0, and 11.0 µM). Inset: The standard curve of the relationship between F_0_/F.

**Table 1 nanomaterials-09-01556-t001:** Excitation wavelength, emission wavelength, quantum yields (QYs), and full width at half maximum (FWHM) of carbon dots (CDs) synthesized by five different solvents.

Solvent	EA	IPA	MT	NBA	DMF
Ex/nm	540	520	480	460	460
Em/nm	605	600	552	543	527
FWHM/nm	56	65	86	75	44
QYs/%	30.2	32.7	10.0	10.0	47.6

**Table 2 nanomaterials-09-01556-t002:** XPS elemental analysis results of the R-CDs and Y-CDs.

Sample	C (mol%)	O (mol%)	N (mol%)
R-CDs	74.45	16.08	9.48
Y-CDs	73.37	5.38	21.25

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
