# Peer review of "Synthesis of Multicolor Carbon Dots Based on Solvent Control and Its Application in the Detection of Crystal Violet"

_nanomaterials, 2019, doi:10.3390/nano9111556_

Round 1

Reviewer 1 Report

In the present manuscript Zhao et al, shows the synthesis a series of Carbon Dots based on solvent control of this process. The authors prove that the different properties of the solvents are fundamental for the photophysical properties of the obtained CDs. Furthermore, the Y-CDs have been proved to be useful in crystal violet (CV) sensing. 

Despite the paper is well written and easy to understand, a few minor points which should be addressed before acceptance.

-Line 71 and 154 space before the references has to be removed

-In the Figure 2, which is the concentration of the measured samples? Furthermore this figure appears before its first call in the main text

-In Fig 3a, how do authors explain the MC long wavelength emission?

-just as a suggestion, shouldn't it be more visual just to put the overlaid emission spectra of the CDs in the different solvents? And as in UV-vis measurements, which are the CDs concentrations used? These data must be added in the text or at least in the Supporting Information file.

-Line 282 Raman instead of Roman

-Figure 6a, the yellow line is difficult to see, I suggest authors to change the color of this graph

-Line 310 FTIR instead of FIRT

-Line 320, controlled instead of control

-Line 321, it makes a mention to section 3.4 which is not in the manuscript

-Figure 7, HOMO instead of HUMO

-In Figure 8 there is a new band centered at 630 nm, do the authors know the origin of this band? 

I would like authors to make a couple of comments about the following questions:

-Do the authors have an explanation for the high steam pressure to be more beneficial for the growth of long wavelength emissive CDs?

-Why are protic solvents beneficial for the growth of long wavelength emissive CDs? 

I think that after revision the paper should be accepted for publication

Author Response

Reviewer #1:

In the present manuscript Zhao et al, shows the synthesis a series of Carbon Dots based on solvent control of this process. The authors prove that the different properties of the solvents are fundamental for the photophysical properties of the obtained CDs. Furthermore, the Y-CDs have been proved to be useful in crystal violet (CV) sensing. 

Despite the paper is well written and easy to understand, a few minor points which should be addressed before acceptance.

Comment 1: Line 71 and 154 space before the references has to be removed.

Response: Thanks for your suggestion. Thanks for your suggestion. Line 71 and 154 space before the references has been removed.

Comment 2: In the Figure 2, which is the concentration of the measured samples? Furthermore this figure appears before its first call in the main text.

Response: We detailed the concentration of the sample in the revised manuscript. ‘As shown in Fig.2, The concentration of the sample to be tested is 1 mg/mL.’ Furthermore, Fig.2 appears in line 203 in the revised manuscript.

Comment 3: In Fig 3a, how do authors explain the MC long wavelength emission?

Response: In our study, acetone (CP), ethyl acetate (EAC) and dichloromethane (MC) are non-protic solvents, but the fluorescence wavelength of prepared CDs in MC is obviously longer than that in other solvents. We think that is because MC molecules contain chlorine atoms, and the existence of lone pairs of chlorine would impact the surface state of carbon dots. The reference to the impact of solvents upon the fluorescence of CDs is quite limited, so further studies need to be done for the researchers.

Comment 4: just as a suggestion, shouldn't it be more visual just to put the overlaid emission spectra of the CDs in the different solvents? And as in UV-vis measurements, which are the CDs concentrations used? These data must be added in the text or at least in the Supporting Information file.

Response: The two figure forms of Figs 3(a) and (b) were compared during our writing of the manuscript. When the wavelength was set as the abscissa, the spectra overlapped greatly because the wavelengths of CDs prepared by some solvents are quite close, while when applying Fig. 3(c) form (UV-vis spectra), the emission spectra were longitudinally compressed and severely deformed. Therefore, these two figure forms both failed to directly express the difference of CDs fluorescence properties prepared by solvents. We thus chose Figs 3(a) and (b) to expresses that.

Comment5: Line 282 Raman instead of Roman

Response: ‘Roman’ has been replaced by ‘Raman’.

Comment 6: Figure 6a, the yellow line is difficult to see, I suggest authors to change the color of this graph.

Response: Thank you for your suggestion. The yellow line has been replaced by the blank line(fig.6).

Figure 6. (a) FTIR spectrum of R-CDs and Y-CDs. (b) N1s spectrum of R-CDs. (c) N1s spectrum of Y-CDs.

Comment7: Line 310 FTIR instead of FIRT.

Response: We have changed ‘FIRT’ to ‘FTIR’.

Comment 8: Line 320, controlled instead of control

Response: This sentence ‘…properties are control by quantum confinement…’has been replaced by ‘…properties are controlled by quantum confinement…’.

Comment 9: Line 321, it makes a mention to section 3.4 which is not in the manuscript

Response: We are so sorry. ‘Section 3.4’ should be modified to ‘section 3.1’.

Comment 10: Figure 7, HOMO instead of HUMO.

Response: We have changed original figure to the revised figure 7.

Figure 7. A possible luminescence mechanism for CDs.

Comment 11: In Figure 8 there is a new band centered at 630 nm, do the authors know the origin of this band?

Response: We are not quite certain about the source of the new peak at 630 nm, so we didn't do a detailed discussion in the manuscript. We proposed the origin of this new peak might be: crystal violet can not only quench the fluorescence of CDs based on inner filter effect, but can also combine with CDs to form complex, which might exhibit weak fluorescence at 630nm. This is our speculation, and more experiments need to be done to verify that.

Comment 12: Do the authors have an explanation for the high steam pressure to be more beneficial for the growth of long wavelength emissive CDs?

Response: It is known that the high steam pressure would set impacts upon the reaction speed and the properties of reactants. For instance, graphene carbon can be turned into high density and ordered diamond state carbon under high pressure environment. During the preparation of CDs, the high steam pressure would press the low intensity reactants to form high intensity polymer, beneficial to the formation of large carbon cores. It can also speed up the reaction process, benefiting the growth of long wavelength emissive CDs.

Comment 13: Why are protic solvents beneficial for the growth of long wavelength emissive CDs? 

Response: Based on the CDs preparation experiments with different solvents, it was discovered that compared with non-protic solvents, protic solvents are more beneficial to the formation of long wavelength emissive CDs. It is well known that water and alcohol solvents (ethanol, isopropanol, methanol and n-butanol) are typical protic solvents. We thought that is because in protic solvents, the hydroxyl group can form hydrogen bonds with amino groups in the carbon source (1,2,4,5-tetraaminobenzene), which could increase conjugate system during the formation of CDs, and more beneficial to the formation of long wavelength emissive CDs. It is worth mentioning is that, when using water, the most typical protic solvent, as solvent, though the fluorescence intensity of prepared CDs is quite weak, its emission wavelength is obviously longer than other solvents. This speculation has been put in the revised manuscript, and we hope that more researchers can join in this discussion.

Reviewer 2 Report

In the first part, the article describes and discusses several routes for the synthesis of carbon dots using different solvents. The impact on the CDs characteristics is carefully presented even if short explanations are given for the difference obtained. Furthermore, it would be interesting to discuss about the reproducibility of each synthesis in terms of characteristics of the obtained CDs. I would suggest integrating more the bare description of the outcome of the experiments with the actual discussion and comments about the results and their meaning.

The paragraph 3.3 should be implemented. The results of this part should be more carefully presented. The experimental procedure for obtaining the detection limit and calibration curve should be clearly presented in “Material and methods” section. It should be discussed in the 3.3 paragraph, comparing with other detection limit presented in the published literature. The relevance of the presented work has to be enlighten, considering the impact on possible application, the importance for the readers and underlying the strong and the weak points.

Author Response

Reviewer #2:

In the first part, the article describes and discusses several routes for the synthesis of carbon dots using different solvents. The impact on the CDs characteristics is carefully presented even if short explanations are given for the difference obtained. Furthermore, it would be interesting to discuss about the reproducibility of each synthesis in terms of characteristics of the obtained CDs. I would suggest integrating more the bare description of the outcome of the experiments with the actual discussion and comments about the results and their meaning.

Comment 1: The paragraph 3.3 should be implemented. The results of this part should be more carefully presented. The experimental procedure for obtaining the detection limit and calibration curve should be clearly presented in “Material and methods” section.

Response: In the paragraph 3.3, we have described it in more detail.‘Fig.8 shows the quenching effect of the fluorescence intensity towards CV, the fluorescence intensity of Y-CDs decreases gradually with increasing concentration of CV(The concentration of the quencher is 0,0.1,0.5, 0.9,1.3,1.8, 2.5,3.0, 4.0, 5.0,6.0,7.0,8.0,9.0,10.0,11.0 μM). The F0/F values of the Y-CDs treated with a concentration gradient of CV. A good linear correlation (R2 =0.99807) was exhibited over the concentration range of 0.1–11μM and the LOD,according to the IUPAC standard, which was taken as 3×standard deviation/slope, was calculated as 20 nM and comparable with the reported data [40].

Comment 2: It should be discussed in the 3.3 paragraph, comparing with other detection limit presented in the published literature.

Response: Chen[1] et al. has realized the detection of crystal violet in fish tissues based on yellow emissive silicon nanoparticles with its detection limit at 25ug/ml (6.12X10-5M), while in our study, the detection limit is 20nm, lower than that in Chen’s detection.  

[1] Han, Y.; Chen, Y.; Liu, J.; Niu, X.; Ma, Y.; Ma, S.; Chen, X. Room-temperature synthesis of yellow-emitting fluorescent silicon nanoparticles for sensitive and selective determination of crystal violet in fish tissues. Sensors and Actuators B: Chemical, 2018,263, 508-516.

Comment 3: The relevance of the presented work has to be enlighten, considering the impact on possible application, the importance for the readers and underlying the strong and the weak points.

Response: For the present, the fluorescence analysis in the detection of crystal violet is quite rare in published literatures. Our detection method is fast, simple with high sensitivity and selectivity. Its defect is that it has been applied in the detection in actual samples.

Reviewer 3 Report

The paper entitled “Synthesis of Multicolor Carbon Dots Based on Solvent Control and its Application in Detection of Crystal Violet” by authors Dan Zhao, et al., discuss the synthesis of carbon dots (CDs) through the change of solvent. The synthesized CDs have red to yellow colour emission and it is possible to utilize them as an fluorescence ink. The effects of the polarity and protic properties of solvents used in syntheses on fluorescence properties are discussed.

The paper seems scientifically sound and the results well presented, therefore, I have only minor comments.

Here I give the list of my comments, suggestions or changes which should be adequately addressed before the paper is published:

1) Ln.47: ‘…seen on published…‘ should be ‘…seen in published…’.

2) Ln.51 and 61: ‘Ding et al.’ and ‘Xiong et al.’ are the same papers (I guess) so they should be cited the same way and in both cases the reference number should be indicated (at ‘Ding et al.’ it is missing).

3) Reference [15] is not cited anywhere in the manuscript.

4) Ln.59: In the ‘Wang and …‘ the reference number should be addressed (I guess [17]).

5) Ln.103-104: ’24 mL’ and ’25 mL’ are these two numbers correct? If so where from the 1 extra mL came from?

6) Ln.110: I don’t understand this sentence: ‘The prepared R-CDs and Y-CDs were added into the column to get the separation solution.’ Especially the part ‘to get the separation solution’. Please reformulate.

7) Ln.114: I don’t understand this sentence: ‘Y-CDs solution, different detected samples and ethanol were made up to 1 mL,…’. Please reformulate.

8) Ln.132: I believe that ‘540-560’ should be ‘460-540’. Please check and fix this.

9) Ln.174: I think reference [13] in ‘Xiong’ s team[13], …’ should be reference [14].

10) Figure 2: I strongly recommend to put both panels from Figure S1 into Figure 2 in the main manuscript (and sort the panels in order they are referenced in the text). It will be much easier for reader to orient.

11) In the whole paper: The word ‘protonic’ should be replaced with ‘protic’.

12) Ln.240-243: The sentence seems bit confusing. Divide it. Also mention (maybe in brackets) which two solvents do you mean?

13) Ln.251-252: ‘…R-CDs exhibit 65nm difference…’ and …’Y-CDs exhibit 251 110nm difference.’ It seem to me that the difference is bigger in both cases.

14) Ln.260: Replace the term ‘quantum size’ with some more accurate descriptor (term).

15) Ln.269: ‘TEM was sued to …’ should be ‘TEM was used to …’.

16) Ln.270: ‘…average particle size at 6.99nm for R-CDs…’. It seems to me that the size is about 9.25 nm?

17a) Ln.274: ‘…(Fig. S3) show peak in 23.1° and 23.2°…’. I cannot find these peaks in the XRD pattern.

17b) Ln.275: ‘This is in consistent with the TEM results that…’. I cannot see any consistency between XRD and TEM. Please clarify 17a) and 17b) more accurately.

18) In Experimental section: Please add how were the FTIR measurements done - which method was used (solution, KBr, ATR)?

19) Ln.321: ‘As discussed in Part 3.4, …’. There is no part 3.4 in the manuscript.

20) Ln.364-365: There should be a bit more discussion about the selectivity. The results indicate that ANY dye detected by Y-CD will “quench” the PL of Y-CD if the absorption spectrum of the dye is overlapping with PL of Y-CD. Some comments of this selectivity problem should be made.

21) There is nowhere in the body of the test mentioned anything about the quantum yields (only in abstract). I think if the authors mention it in the abstract there should also be a paragraph about QY in the body of the paper.

Author Response

Reviewer #3:

The paper entitled “Synthesis of Multicolor Carbon Dots Based on Solvent Control and its Application in Detection of Crystal Violet” by authors Dan Zhao, et al., discuss the synthesis of carbon dots (CDs) through the change of solvent. The synthesized CDs have red to yellow colour emission and it is possible to utilize them as an fluorescence ink. The effects of the polarity and protic properties of solvents used in syntheses on fluorescence properties are discussed.

The paper seems scientifically sound and the results well presented, therefore, I have only minor comments.

Here I give the list of my comments, suggestions or changes which should be adequately addressed before the paper is published:

Comment 1: Ln.47: ‘…seen on published…‘ should be ‘…seen in published…’.

Response: Thank you for you remind. ‘…seen on published…’ has been modified to ‘…seen in published…’.

Comment 2:Ln.51 and 61: ‘Ding et al.’ and ‘Xiong et al.’ are the same papers (I guess) so they should be cited the same way and in both cases the reference number should be indicated (at ‘Ding et al.’ it is missing).

Response: We have changed ‘Ding et al. realized the adjustment of CDs…’to ‘Xiong et al.[14] realized the adjustment of CDs…’.

Comment 3: Reference [15] is not cited anywhere in the manuscript.

Response: The reference [15] was addressed. ‘Tian [15] and his group also realized all color emissive CDs through three different solvents (water, glycerin and N, N-dimethylformamide) with citric acid and urea as raw materials’.

Comment 4:  Ln.59: In the ‘Wang and … ‘the reference number should be addressed (I guess [17]).

Response: The reference 17 was addressed. ‘Wang[17] and his team proposed that the absorption and photoluminescence of CDs in alcoholic solvents are mainly determined by the –OH groups of solvents rather than their polarity’.

Comment 5:  Ln.103-104: ‘24 mL’ and ‘25 mL’ are these two numbers correct? If so where from the 1 extra mL came from?

Response: Thank you for your question. We have modified ‘25 mL solution was then transferred into a stainless-steel autoclave with Teflon lining’ to ‘24 mL solution was then transferred into a stainless-steel autoclave with Teflon lining’.

Comment 6: Ln.110: I don’t understand this sentence: ‘The prepared R-CDs and Y-CDs were added into the column to get the separation solution.’ Especially the part ‘to get the separation solution’. Please reformulate.

Response: We have changed ‘The prepared R-CDs and Y-CDs were added into the column to get the separation solution.’ to ‘The prepared R-CDs and Y-CDs were added into the column to be separated. Then different separated components can be obtained for characterization.’ So that readers can understand more.

Comment 7: Ln.114: I don’t understand this sentence: ‘Y-CDs solution, different detected samples and ethanol were made up to 1 mL,…’. Please reformulate.

Response: ‘Y-CDs solution, different detected samples and ethanol were made up to 1 mL,…’was modified to ‘The total volume of Y-CDs solution, different detected samples and ethanol is 1ml,…’.

Comment 8:  Ln.132: I believe that ‘540-560’ should be ‘460-540’. Please check and fix this.

Response: Thank you very much for your correction. We modified ‘Their excitation wavelengths are in the range of 540-560 nm,…’ to ‘Their excitation wavelengths are in the range of 460-540 nm, ….’.

Comment 9: Ln.174: I think reference [13] in ‘Xiong’ s team[13], …’ should be reference [14].

Response: We modified ‘…the reports of Xiong’ s team[13],…’ to‘…the reports of Xiong’ s team[14],…’.

Comment 10: Figure 2: I strongly recommend to put both panels from Figure S1 into Figure 2 in the main manuscript (and sort the panels in order they are referenced in the text). It will be much easier for reader to orient.

Response: Thanks for your suggestion. We have moved the Figure S1 to Figure 2 in revised manuscript.

Comment 11: In the whole paper: The word ‘protonic’ should be replaced with ‘protic’.

Response: We have listed the corrected locations as follows.

A list of changes

Original manuscript

Revised manuscript

…protonic and non-protonic property…

…protic and non-protic property…

…their protonic property…

…their protic property…

…polar protonic solutions…

…polar protic solutions…

…polar non-protonic solution…

…polar non-protic solution…

…For protonic solvents…

…For protic solvents…

…some alcoholic protonic solvents…

…some alcoholic protic solvents…

…decrease of protonic solvents...

…decrease of protic solvents…

…interaction between protonic solution and CDs…

…interaction between protic solution and CDs…

For non-protonic solvents…

For non-protic solvents…

In different non-protonic solvents…

In different non-protic solvents…

Comment 12:  Ln.240-243: The sentence seems bit confusing. Divide it. Also mention (maybe in brackets) which two solvents do you mean?

Response: Thanks for your suggestion. We have noted the types of two solvents in the text. —‘…with these two solvents(ethanol and n-butanol),…’. In order not to confuse the reader, we have modified the sentence and modified it as follows: ‘Some means of characterization were used to characterize the CDs as-prepared with these two solvents (ethanol and n-butanol), and further compared their difference in particle size and surface component.’

Comment 13: Ln.251-252: ‘…R-CDs exhibit 65nm difference…’ and …’Y-CDs exhibit 251 110nm difference.’ It seem to me that the difference is bigger in both cases.

Response: The figure on the left (Fig.S1) shows the fluorescence spectra of separation products of R-CDs, and the figure on the right (Fig.4c) is the spectra of separation products of Y-CDs, which shows obvious difference in emission wavelength.

Comment 14: Ln.260: Replace the term ‘quantum size’ with some more accurate descriptor (term).

Response: Thank you for your suggestion. ‘…The size effect is the result of quantum size,…’has been modified to‘’

Comment 15: Ln.269: ‘TEM was sued to …’ should be ‘TEM was used to …’.

Response: ‘TEM was sued to …’was corrected to ‘TEM was used to …’.

Comment 16:  Ln.270: ‘…average particle size at 6.99 nm for R-CDs…’. It seems to me that the size is about 9.25 nm?

Response: We are sorry for this typo. We have correct the sentence “…average particle size at 6.99 nm for R-CDs…” into “…average particle size at 9.39 nm for R-CDs…”.

Comment 17a: Ln.274: ‘…(Fig. S3) show peak in 23.1° and 23.2°…’. I cannot find these peaks in the XRD pattern.

Response: Thanks for your suggestion. We checked Fig. S3 again, and the prepared CDs are mainly amorphous carbon without good crystal structure, it cannot exhibit obvious peaks. So we have removed relevant data in the revised manuscript.

Comment 17b:  Ln.275: ‘This is in consistent with the TEM results that…’. I cannot see any consistency between XRD and TEM. Please clarify 17a) and 17b) more accurately.

Response: Thank you for your suggestion. We have removed this sentence (‘This is in consistent with the TEM results that…’).

Comment 18: In Experimental section: Please add how were the FTIR measurements done - which method was used (solution, KBr, ATR)?

Response: We have added the method we have adopted in experimental section. --‘Fourier infrared spectrometer were purchased from Thermo Fisher Scientific ( KBr tableting method), USA’.

Comment 19: Ln.321: ‘As discussed in Part 3.4, …’. There is no part 3.4 in the manuscript.

Response: We are so sorry. ‘As discussed in Part 3.4, …’was modified to ‘As discussed in Part 3.1, …’.

Comment 20: Ln.364-365: There should be a bit more discussion about the selectivity. The results indicate that ANY dye detected by Y-CD will “quench” the PL of Y-CD if the absorption spectrum of the dye is overlapping with PL of Y-CD. Some comments of this selectivity problem should be made.

Response: For the detection methods based on inner filter effect, the realization of sensitive detection not only relies on the overlapping extent of fluorescence spectra of CDs with absorption spectra of crystal violet, but also on the distance between the energy donor (Y-CDs) and its acceptor (dye). We don’t think that any overlapping of spectra could be applied to the dye detection, but some other requirements need to be meet. First, the overlapping extent varies for different dyes. Only the high overlapping extent can possibly realize the sensitive detection of dyes. Second, to ensure the energy send from Y-CDs can be efficiently absorbed by dye molecules, the distance between the energy donor and the acceptor should be short enough (2~5nm). Therefore, this method cannot be applied to the dyes that show repulsive effect with CDs. Third, the fluorescence probe and dye should exhibit excellent stability and solubility in the given detection solvent. These three requirements have been added into the revised manuscript.

Comment 21: There is nowhere in the body of the test mentioned anything about the quantum yields (only in abstract). I think if the authors mention it in the abstract there should also be a paragraph about QY in the body of the paper.

Response: In addition to the QYs appearing in the abstract, it also appeared in the line 141 the body of paper. We list the QYs data in Table 1. We have moved the Quantum yield (QY) measurement method in the supplementary material of the original manuscript to part 2.4(the Quantum yield (QY) measurement) in the revised manuscript.

‘Rhodamine 6G (QY = 95%) and Rhodamine B(QY = 56%) dissolved in absolute ethanol (refractive index η = 1.096, 25 ° C) were used as a reference[24]. dissolved in absolute ethanol was used as a reference. The QY of CDs was calculated by comparing the ratio of the fluorescence area (Rhodamine 6G-λex = 488 nm and Rhodamine B-λex = 495 nm) to the absorbance. All samples were dissolved in absolute ethanol and their absorbance at 488 nm or 495 nm was controlled to be less than 0.1. The relative QY was calculated as follows:

                        ΦX=ΦST (GradX/ GradST)( ηX2/ηST2)

Φ is QY, Grad is the ratio of the fluorescence area to the absorbance, η is the refractive index of the solvent, ST is the standard substance, and X is the QDs sample, where ηX = ηST.’
